# S3M3D: Cross-Modal Selective State Space Modulation for 3D Object Detection

## Abstract

Multi-modal 3D object detection is critical for autonomous driving. However, prevailing query-based methods suffer from a symmetric fusion bottleneck, treating geometrically precise LiDAR queries and uncertain camera queries with equal reliability. This overlooks the opportunity to use high-fidelity LiDAR queries to guide the interpretation of noise-prone camera queries. To address this, we propose Selective State Space Modulation for 3D object detection (S3M3D), a novel framework applying two synergistic and Mamba-based components to redefine intra- and inter-modality interactions. First, we introduce Spatially-Aware Mamba (SA-Mamba) to model interactions among LiDAR queries, replacing self-attention. It efficiently captures geometric priors by leveraging recursive Z-order serialization of their projected BEV coordinates. Second, we propose LiDAR-Guided Mamba (LG-Mamba) to establish an asymmetric guidance mechanism, where the robust LiDAR queries dynamically modulate the state-space processing of the less reliable camera queries. This allows geometric structure to actively steer semantic feature refinement. Extensive experiments on nuScenes demonstrate that S3M3D achieves state-of-the-art performance.

## 1 Introduction

Autonomous driving systems critically rely on robust 3D object detection to perceive their environment. However, achieving such robustness with a single sensing modality remains difficult (Wang et al., 2023b;c). LiDAR excels at geometry but degrades in adverse weather (Yin et al., 2021; Wang et al., 2023a; Huang et al., 2025), while cameras provide rich semantics yet struggle with depth estimation (Han et al., 2022; Yan et al., 2024; Wu et al., 2024). This complementarity establishes multi-modal fusion as a dominant paradigm, centering the research challenge on cross-modal interaction design (Liang et al., 2022; Bai et al., 2022; Liu et al., 2023; Chen et al., 2023).

Many existing methods address multi-modal fusion by unifying modalities in the shared BEV space (Liang et al., 2022; Liu et al., 2023). Unlike LiDAR point clouds, which can be readily voxelized into a high-fidelity BEV feature map, the camera modality requires a more complex 2D-to-3D lifting process that projects image features into 3D space through depth estimation (Huang et al., 2021; Li et al., 2022b). The resulting camera-BEV representation often suffers from geometric distortions and uncertainties (Zhou et al., 2023; Li et al., 2022a). Consequently, directly fusing this potentially noisy camera-BEV with the precise LiDAR-BEV creates a fundamental bottleneck as errors from the vision modality can degrade the final fused representation (Liang et al., 2022; Liu et al., 2023).

To overcome the limitations of BEV-level fusion, a new paradigm based on object queries has emerged, exemplified by DETR3D (Wang et al., 2021) and FUTR3D (Chen et al., 2023). While these methods bypass the explicit BEV construction for cameras, they introduce two overlooked challenges in how queries interact. First, for intra-modality interaction, vanilla self-attention for LiDAR queries is ill-suited. Its permutation invariance prevents it from capturing the queries' spatial arrangement and crucial geometric priors, such as the strong correlation between neighboring queries. Second, for inter-modality fusion, a fundamental difference exists in query reliability: LiDAR queries are geometrically precise, whereas camera queries are prone to projection noise. Previous methods ignore this, using symmetric mechanisms that treat both as equal partners and fail to leverage high-fidelity LiDAR queries to guide noisier camera queries. This dual bottleneck of structure-agnostic modeling and symmetric fusion significantly degrades detection performance.

To address these challenges, we introduce Selective State Space Modulation for 3D object detection (S3M3D), a novel architecture advancing both intra-modality query interaction and inter-modality query fusion. First, to address the structure-agnostic limitations of self-attention, we introduce the Spatially-Aware Mamba (SA-Mamba) module. This module serializes queries based on their spatial locality using a recursive Z-order curve, enabling the Mamba architecture to capture crucial geometric priors and establish a more context-aware LiDAR representation. Second, to overcome the symmetric fusion bottleneck, we propose the LiDAR-Guided Mamba (LG-Mamba) module. This

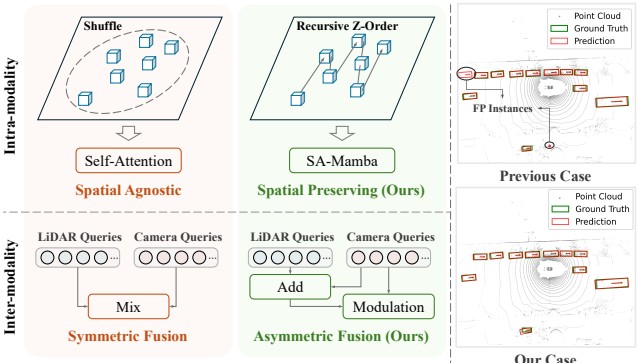

Figure 1: Our method improves detection with two designs: 1) a spatial-preserving SA-Mamba modeling LiDAR queries along a recursive BEV Z-order curve, and 2) asymmetric LiDAR-guided inter-modality fusion. Right: qualitative comparison shows FP elimination.

module uses the refined LiDAR queries from SA-Mamba to dynamically modulate the state-space processing of less reliable camera queries, establishing a robust, guidance-based fusion mechanism resilient to projection uncertainties. Crucially, the two modules are co-designed: SA-Mamba yields high-fidelity LiDAR query representations that LG-Mamba uses to modulate the state-space processing of camera queries, yielding robust, LiDAR-guided cross-modal fusion and more geometrically grounded multi-modal perception. Figure 1 shows SA-Mamba and LG-Mamba replacing spatially agnostic self-attention and symmetric fusion, improving 3D detection accuracy.

Our main contributions are summarized as follows:

- We introduce SA-Mamba, a geometry-aware module for LiDAR query interaction. It preserves the BEV spatial locality by reordering queries along a recursive Z-order curve, making it a powerful replacement for self-attention in intra-modality modeling.

- We propose LG-Mamba, a cross-modal fusion mechanism that departs from the paradigm of symmetric fusion. It asymmetrically modulates the state-space processing of camera queries using LiDAR queries, creating a guidance pathway from LiDAR to camera.

- We integrate these components into a unified framework named S3M3D and validate its effectiveness through extensive experiments on the nuScenes benchmark. The state-of-the-art results underscore the importance of the synergistic design.

## 2 Related Work

### 2.1 Multi-modal 3D Object Detection

Fusing complementary LiDAR and camera data is crucial to modern 3D perception. While early methods like PointPainting (Vora et al., 2020) operated at the point level, the field has shifted towards two dominant feature-level fusion paradigms. The first is BEV-level fusion, where methods like BEVFusion (Liang et al., 2022; Liu et al., 2023) project camera features into the Bird's-Eye-View (BEV) for direct alignment with LiDAR features. More recent approaches like UVTR (Li et al., 2022a) seek an even tighter integration by establishing a unified voxel representation. The second is the query-based paradigm. Pioneered by DETR3D (Wang et al., 2021) and extended to multi-modal settings by models like TransFusion (Bai et al., 2022) and FUTR3D (Chen et al., 2023), this approach uses a unified set of object queries to interact with both LiDAR and camera features. Similarly, CMT (Yan et al., 2023) employs a dedicated cross-modal Transformer to deeply integrate features from both sensors. Along this line, DeepInteraction (Yang et al., 2022) and DeepInteraction++ (Yang et al., 2025) introduce a graph-based approach to facilitate early interaction between modal-specific queries, aiming for a more tightly coupled representation before feature sampling.

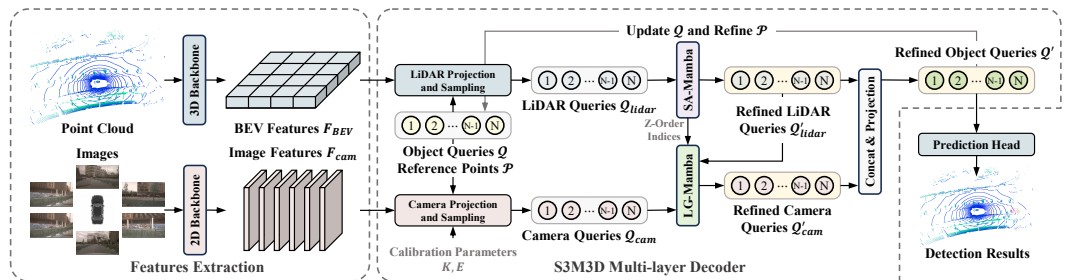

Figure 2: Overall architecture of our proposed S3M3D. The model takes LiDAR point clouds and multi-view images as the input. Each learnable object query is associated with a 3D reference point and subsequently processed in a multi-layer decoder. In each layer, LiDAR queries are first structured by SA-Mamba to capture geometric locality and then serve as a conditioning signal in LG-Mamba to modulate the processing of camera queries, enabling asymmetric cross-modal interaction.

## 2.2 STATE SPACE MODELS IN VISION

State Space Models (SSMs), particularly Mamba (Gu & Dao, 2023), have emerged as an efficient alternative to Transformers (Vaswani et al., 2017). By replacing quadratic self-attention with a linear-time selective scan, SSMs offer compelling scalability. Adapting these inherently 1D models to vision requires techniques to flatten spatial features into sequences while preserving locality, such as the orthogonal and bidirectional scanning used in VMamba (Liu et al., 2024a). In 3D object detection, this trend has focused on designing LiDAR backbones. Strategies include group-free methods like Voxel-Mamba (Zhang et al., 2024b), which serializes all voxels using a Hilbert curve, and window-based methods like LION (Liu et al., 2024b), which processes local voxel groups. While these works validate SSMs for dense, voxel-level feature encoding, their potential has remained confined to the backbone. The application of SSMs to model interactions among sparse object queries and facilitate sophisticated asymmetric cross-modal fusion remains largely unexplored.

## 3 METHOD

### 3.1 PRELIMINARIES

**Query-based Multi-modal 3D Object Detection.** Modern 3D detectors (Chen et al., 2023) employ a set of learnable object queries $\mathcal{Q} = \{q_i \in \mathbb{R}^C\}_{i=1}^N$, each associated with a 3D reference point $\mathcal{P} = \{p_i \in \mathbb{R}^3\}_{i=1}^N$. These query-point pairs are iteratively refined within a multi-layer decoder. First, each query samples features from sensor-specific maps guided by its reference point.

For the LiDAR branch, deformable attention (Zhu et al., 2021) is used. It predicts sampling offsets $\{\Delta p_{ik}\}$ and attention weights $\{w_{ik}\}$ from the query $q_i$ to sample the BEV feature map $F_{bev}$:

$$q_{lidar,i} = \sum_{k=1}^{K} w_{ik} \cdot \text{Sample}(F_{bev}, (p_{i,x}, p_{i,y}) + \Delta p_{ik}), \tag{1}$$

where $K$ is the number of sampling points per query in the LiDAR branch.

For the camera branch, the 3D reference point $p_i$ is projected onto each image view $j$ to obtain pixel coordinates $(u_{ij}, v_{ij}) = \Pi(p_i, K_j, E_j)$. Camera queries are then sampled from the multi-view camera feature map $F_{cam}$:

$$q_{cam,i} = \sum_{j=1}^{M} \mathcal{V}(p_i, j) \cdot \text{Sample}(F_{cam,j}, (u_{ij}, v_{ij})), \tag{2}$$

where $F_{cam,j}$ denotes the $j$-th view of the camera feature map, $\mathcal{V}(p_i, j)$ is a visibility indicator that indicates whether the reference point $i$ is projected within the camera plane of view $j$, and $M$ is the number of camera views.

Following feature aggregation, self-attention is first applied to the LiDAR queries $\mathcal{Q}_{lidar} = \{q_{lidar,i} \in \mathbb{R}^{C_l}\}_{i=1}^N$ to model their inter-relationships. These refined LiDAR queries are then element-wise fused with the corresponding camera queries $\mathcal{Q}_{cam} = \{q_{cam,i} \in \mathbb{R}^{C_c}\}_{i=1}^N$. The resulting fused representation is then used to update the object queries $\mathcal{Q}$ and predict corresponding offsets $\{\Delta p_i\}$ to refine the reference points $\mathcal{P}$ ($p_i \leftarrow p_i + \Delta p_i$).

**State Space Models (SSMs).** SSM is a powerful architecture for sequence modeling, governed by a linear ordinary differential equation (ODE):

$$h'(t) = \mathbf{A}h(t) + \mathbf{B}x(t), \quad y(t) = \mathbf{C}h(t) + \mathbf{D}x(t), \tag{3}$$

where $x(t)$ and $y(t)$ are the input and output, $h(t)$ is a latent state, and $\mathbf{A}, \mathbf{B}, \mathbf{C}, \mathbf{D}$ are parameter matrices. For discrete sequences, the system is discretized with a timescale parameter $\Delta$ denoted as:

$$h_k = \overline{\mathbf{A}}h_{k-1} + \overline{\mathbf{B}}x_k, \quad y_k = \mathbf{C}h_k + \mathbf{D}x_k. \tag{4}$$

The discrete matrices $\overline{\mathbf{A}}$ and $\overline{\mathbf{B}}$ are derived from their continuous counterparts via the zero-order hold (ZOH) rule:

$$\overline{\mathbf{A}} = e^{\Delta\mathbf{A}}, \quad \overline{\mathbf{B}} = (\Delta\mathbf{A})^{-1}(e^{\Delta\mathbf{A}} - \mathbf{I})\Delta\mathbf{B}. \tag{5}$$

Mamba makes $\Delta$, $\mathbf{B}$, and $\mathbf{C}$ input-dependent, enabling content-aware sequence modeling.

## 3.2 Overall Architecture

As depicted in Figure 2, our S3M3D framework processes LiDAR and image inputs through standard backbones to generate feature maps. The core of our method is a multi-layer decoder that iteratively refines a set of object queries. Within each decoder layer, an asymmetric refinement process occurs. First, the SA-Mamba module structures the LiDAR queries by their BEV locality to capture geometric context. These refined LiDAR queries then act as a conditioning signal for our LG-Mamba module, which asymmetrically modulates the processing of the corresponding camera queries. The updated queries from both modalities are then aggregated and passed to the next layer. After the final layer, the query representations are fed into the prediction head to produce 3D bounding boxes and class labels.

## 3.3 Spatially-Aware Mamba

The inherent limitation of self-attention in query-based 3D object detectors is that it treats object queries as an unstructured set, ignoring their crucial spatial relationships. To address this, we introduce SA-Mamba. It restructures the query interaction by leveraging a spatially-aware processing order, thereby enabling the task to benefit from the efficiency and inductive biases of SSMs. This is achieved by ordering the queries based on their BEV spatial locality, allowing the sequential architecture of Mamba to explicitly capture geometric context.

Given the LiDAR queries $\mathcal{Q}_{lidar} = \{q_{lidar,i} \in \mathbb{R}^C\}_{i=1}^N$ and their corresponding 3D reference points $\mathcal{P} = \{p_i \in \mathbb{R}^3\}_{i=1}^N$, the SA-Mamba layer operates in three stages:

**Recursive Z-order Serialization.** As depicted in Figure 3 (a), the process begins by projecting the 3D reference points $\mathcal{P}$ onto the BEV plane. To impose spatial order and preserve locality, we leverage the recursive Z-order curve. The recursive nature of this method illustrated by the magnified view in Figure 3 (c) ensures that the space is continuously subdivided until

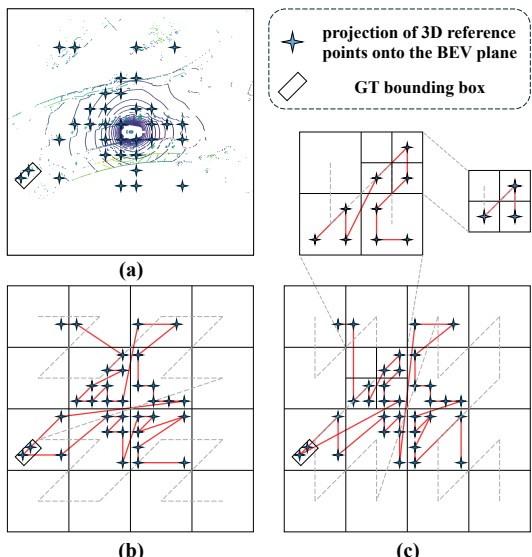

Figure 3: SA-Mamba transforms object queries into structured sequences using recursive Z-order curve. (a) 3D reference points are projected onto the BEV plane. (b, c) Orthogonal Z-order scans are applied, generating query sequences, where recursive subdivision ensures each grid contains at most one reference point.

each grid cell contains at most one reference point. We define the serialization by sorting the Z-order indices of the BEV coordinates $(u_i, v_i)$, yielding a permutation $\pi_x$.

$$\pi_x = \text{argsort}(\{\mathcal{Z}(u_i, v_i)\}_{i=1}^N), \tag{6}$$

where $\mathcal{Z}(\cdot, \cdot)$ is the Z-order function that computes the Z-order values at the finest grid level resulting from the recursive subdivision.

Applying the indices $\pi_x$ to reorder the queries yields a spatially sorted sequence $\mathcal{Q}_{lidar}^x = \mathcal{Q}_{lidar}[\pi_x]$, whose scan path is visualized in Figure 3 (b).

**Bidirectional Context Aggregation.** To ensure each query aggregates information from its entire neighborhood along the sequence, we process the $\mathcal{Q}_{lidar}^x$ bidirectionally. This is achieved by applying the Mamba block in both the forward direction and the backward direction. We define this entire operation as a single block $\mathcal{B}$:

$$\mathcal{B}(\mathcal{Q}_{lidar}^x) = \mathcal{M}(\mathcal{Q}_{lidar}^x) + \mathcal{R}(\mathcal{M}(\mathcal{R}(\mathcal{Q}_{lidar}^x))), \tag{7}$$

where $\mathcal{M}$ denotes the Mamba block (Gu & Dao, 2023) processed by Equation 3, 4, and 5. $\mathcal{R}$ denotes the sequence reversal operation.

**Isotropic Representation via Orthogonal Scans.** A single scan direction can introduce directional bias. To create a more robust and isotropic query representation, we complement the primary scan $\pi_x$ with an orthogonal one $\pi_y$, generated by transposing the BEV coordinates. This creates a complementary query sequence:

$$\pi_y = \text{argsort}(\mathcal{Z}(v_i, u_i)_{i=1}^N), \mathcal{Q}_{lidar}^y = \mathcal{Q}_{lidar}[\pi_y]. \tag{8}$$

Each sequence is processed independently by the bidirectional context aggregation block. The resulting feature sets are then remapped to their original query order and fused via element-wise addition to produce the final refined LiDAR queries $Q_{lidar}'$:

$$\mathcal{Q}_{lidar}' = \mathcal{U}(\mathcal{B}(\mathcal{Q}_{lidar}^x), \pi_x) + \mathcal{U}(\mathcal{B}(\mathcal{Q}_{lidar}^y), \pi_y), \tag{9}$$

where $\mathcal{U}(\cdot, \pi)$ denotes the un-shuffling operator that restores a sequence to its order before being rearranged by $\pi$. This multi-scan interaction ensures the learned spatial relationships are direction-invariant. The refined LiDAR queries $Q_{lidar}'$ encode geometric priors.

To enable deeper geometric reasoning, we construct the SA-Mamba module by stacking multiple layers described above. A crucial design principle is that the initial Z-order permutations $(\pi_x, \pi_y)$, computed once from the reference points, are reused across all subsequent layers. This establishes a fixed, spatially-coherent processing path for the queries. As LiDAR query features are passed hierarchically from one layer to the next, this consistent ordering allows the model to build upon previously aggregated context. This iterative refinement enables the capture of increasingly complex geometric patterns, leading to a more robust and sophisticated final representation for each query.

### 3.4 LiDAR-Guided Mamba

A primary challenge in multi-modal fusion is that existing methods treat the LiDAR and camera queries as equally reliable. This simplistic assumption allows the spatial ambiguity inherent in camera projections to corrupt the geometrically precise LiDAR queries. Our approach recognizes that while camera queries offer rich semantics, LiDAR queries provide a more trustworthy structural foundation. Therefore, we propose LG-Mamba, an asymmetric fusion module that leverages the high-fidelity LiDAR queries to guide the processing of camera queries. This strategy effectively utilizes the semantic richness of camera queries while mitigating the risks of their spatial uncertainty.

The architecture of our proposed module is depicted in Figure 4. The overall LG-Mamba layer, shown in Figure 4 (a), employs a bidirectional structure. It processes the input query sequences ($\mathcal{Q}_{cam}$ and $\mathcal{Q}_{lidar}'$) in both a forward direction and a backward direction, with the results aggregated via element-wise addition. The core innovation resides in the LG-Mamba block, detailed in Figure 4 (b). Here, the sorted camera and LiDAR queries are first projected. A unified modulation signal is created by combining both projected queries to generate the dynamical parameters ($\Delta, \mathbf{B}, \mathbf{C}$)

for SSM. Crucially, the camera query stream alone serves as the primary input. This mechanism ensures that LiDAR's structural information guides the processing of camera queries without direct feature-level corruption.

**Structurally-Aligned Co-Serialization.** The cornerstone of LG-Mamba is imposing the authoritative geometric order from LiDAR onto the camera modality. We reuse the permutations $\pi_x$ and $\pi_y$ to co-serialize both the LiDAR and camera query sets into two spatially-meaningful sequences:

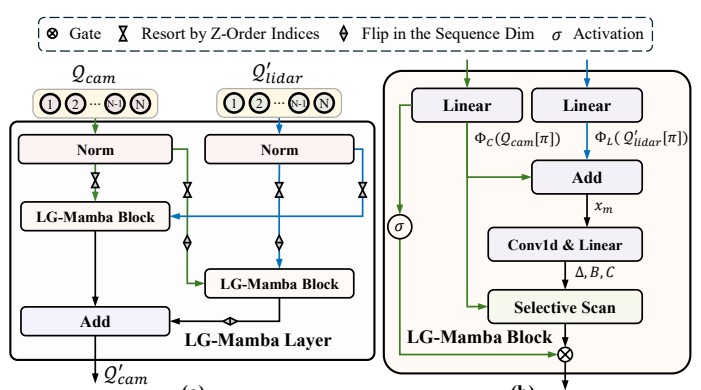

$$\begin{aligned} Q'^{x}_{lidar} &= Q'_{lidar}[\pi_x], \\ Q^{x}_{cam} &= Q_{cam}[\pi_x], \\ Q'^{y}_{lidar} &= Q'_{lidar}[\pi_y], \\ Q^{y}_{cam} &= Q_{cam}[\pi_y]. \end{aligned} \quad (10)$$

This operation aligns the LiDAR queries and camera queries, establishing the foundation for guided fusion along two complementary paths.

Figure 4: Architecture of the LG-Mamba module. (a) The module consists of $L$ stacked LG-Mamba layers. Each layer employs a bidirectional Mamba structure, processing the sequence in both forward and backward directions. (b) In the LG-Mamba Block, robust LiDAR queries are integrated with origin camera queries to generate modulation cues that determine the $(\Delta, \mathbf{B}, \mathbf{C})$ of the Mamba model processing the camera queries.

For clarity and conciseness, we will now describe the core modulation mechanism using a single, generic permutation $\pi$, which can represent either $\pi_x$ or $\pi_y$. The process described is applied independently to both the $x$-aligned and $y$-aligned sequences.

**LiDAR-Guided State-Space Modulation.** With the queries aligned, we implement an asymmetric guidance mechanism. The process begins by projecting the LiDAR and camera queries and resulting a single modulation signal $x_m$:

$$x_m = x_c + x_l = \Phi_C(Q_{cam}[\pi]) + \Phi_L(Q'_{lidar}[\pi]), \quad (11)$$

where $\Phi_C$ and $\Phi_L$ are linear layers.

Next, this modulation signal $x_m$ is passed through a lightweight network to dynamically generate the input-dependent SSM parameters $(\Delta, \mathbf{B}, \mathbf{C})$:

$$[\Delta, \mathbf{B}, \mathbf{C}] = \Phi_P(\text{SiLU}(\text{Conv1D}(x_m))). \quad (12)$$

where $\Phi_P$ denotes linear layers.

Crucially, these parameters are then used to steer the state-space model's processing of the original camera stream $x_c$. This ensures that the modulation signal $x_m$ dynamically provides high-level guidance for camera query updates.

Specifically, the state transition of the camera query sequence is governed by the discrete SSM equation (Equation 4), where the input sequence $\{x_k\}$ is the camera query stream $x_c$. However, the state matrices $\overline{\mathbf{A}}$ and $\overline{\mathbf{B}}$ derived from $\Delta$ and $\mathbf{B}$ (Equation 5) and the output matrix $\mathbf{C}$ are dynamically computed from the combined modulation signal $x_m$. This mechanism, in which we term LiDAR-Guided Scan as $\mathcal{G}(Q_{cam}, Q'_{lidar})$, ensures that the structural information from LiDAR queries dictates how the semantic information within camera queries is processed and propagated, without directly corrupting the camera features themselves.

**Bidirectional Fusion and Orthogonal Aggregation.** To build a comprehensive and isotropic representation, we process the two orthogonal sequences bidirectionally and fuse the results. First, we

Table 1: Performance comparison on the nuScenes 3D detection dataset. All results are from single models without test-time augmentation (TTA) or ensembling. For each dataset split (validation/test), the **best** mAP and NDS scores for LiDAR-only (L) and LiDAR-Camera (LC) models are in bold. 'C.V.', 'Motor.', 'Ped.' and 'T.C.' are short for construction vehicle, motorcycle, pedestrian, and traffic cones. '-' indicates that the results are not publicly available.

| Methods | Modality | mAP | NDS | Car | Truck | C.V. | Bus | Trailer | Barrier | Motor. | Bike | Ped. | T.C. |
|---|---|---|---|---|---|---|---|---|---|---|---|---|---|
| **Performances on the validation set** | | | | | | | | | | | | | |
| CenterPoint (Yin et al., 2021) | L | 56.9 | 65.3 | 85.0 | 53.8 | 16.4 | 66.5 | 33.1 | 68.2 | 55.9 | 37.7 | 84.4 | 68.1 |
| SAFDNet (Zhang et al., 2024a) | L | 66.3 | 71.0 | 87.6 | 60.8 | 26.6 | 78.0 | 43.5 | 69.7 | 75.5 | 58.0 | 87.8 | 75.0 |
| BEVFusion (Liu et al., 2023) | LC | 67.9 | 71.0 | 88.6 | 65.0 | 28.1 | 75.4 | 41.4 | 72.2 | 76.7 | 65.8 | 88.7 | 76.9 |
| BEVFusion (Liang et al., 2022) | LC | 69.6 | 72.1 | 89.1 | 66.7 | 30.9 | 77.7 | 42.6 | 73.5 | 79.0 | 67.5 | 89.4 | 79.3 |
| UniPAD (Yang et al., 2024) | LC | 69.9 | 73.2 | - | - | - | - | - | - | - | - | - | - |
| VirPNet (Wang et al., 2024) | LC | 70.4 | 73.2 | 87.9 | 60.0 | 30.1 | 67.9 | 58.6 | 76.3 | 70.9 | 50.8 | 90.3 | 85.9 |
| FUTR3D-L (Chen et al., 2023) | L | 63.7 | 69.1 | 85.9 | 55.7 | 27.2 | 75.6 | 44.8 | 64.2 | 72.5 | 54.8 | 84.3 | 71.5 |
| S3M3D-L (Ours) | L | **67.0** | **71.3** | 88.3 | 61.6 | 29.7 | 75.3 | 48.7 | 67.8 | 74.9 | 59.4 | 87.1 | 77.2 |
| FUTR3D (Chen et al., 2023) | LC | 70.3 | 73.2 | 88.4 | 67.2 | 33.8 | 78.7 | 46.9 | 71.3 | 79.8 | 72.2 | 86.6 | 78.4 |
| S3M3D (Ours) | LC | **71.1** | **73.7** | 89.8 | 68.4 | 32.3 | 77.1 | 51.7 | 71.7 | 80.8 | 72.3 | 87.9 | 79.0 |
| **Performances on the test set** | | | | | | | | | | | | | |
| TransFusion (Bai et al., 2022) | LC | 68.9 | 71.7 | 87.1 | 60.0 | 33.1 | 68.3 | 60.8 | 78.1 | 73.6 | 52.9 | 88.4 | 86.7 |
| BEVFusion (Liu et al., 2023) | LC | 69.2 | 71.8 | 88.1 | 60.9 | 34.4 | 69.3 | 62.1 | 78.2 | 72.2 | 52.2 | 89.1 | 85.2 |
| BEVFusion (Liang et al., 2022) | LC | 70.2 | 72.9 | 88.6 | 60.1 | 39.3 | 63.8 | 69.8 | 80.0 | 74.1 | 51.0 | 89.2 | 86.5 |
| FUTR3D-L (Chen et al., 2023) | L | 65.3 | 69.9 | - | - | - | - | - | - | - | - | - | - |
| S3M3D-L (Ours) | L | **67.8** | **71.8** | 87.4 | 56.6 | 34.0 | 69.1 | 62.3 | 73.5 | 76.3 | 48.2 | 86.3 | 84.2 |
| FUTR3D (Chen et al., 2023) | LC | 69.4 | 72.1 | - | - | - | - | - | - | - | - | - | - |
| S3M3D (Ours) | LC | **71.2** | **73.6** | 87.9 | 61.0 | 34.9 | 66.5 | 71.4 | 79.2 | 81.2 | 56.4 | 88.7 | 85.0 |

define a bidirectional LG-Mamba block, $\mathcal{B}_{LG}$, which applies the guided scan in both forward and reverse directions:

$$\mathcal{B}_{LG}(\mathcal{Q}_{cam}, \mathcal{Q}'_{lidar}) = \mathcal{G}(\mathcal{Q}_{cam}, \mathcal{Q}'_{lidar}) + \mathcal{R}(\mathcal{G}(\mathcal{R}(\mathcal{Q}_{cam}), \mathcal{R}(\mathcal{Q}'_{lidar}))). \tag{13}$$

This block is then applied independently to the $x$-aligned and $y$-aligned query pairs. The resulting sequences are un-shuffled back to their original order using the inverse permutations and fused via element-wise addition to produce the final refined camera queries $\mathcal{Q}'_{cam}$:

$$\begin{aligned} Y^x_{cam} &= \mathcal{B}_{LG}(\mathcal{Q}^x_{cam}, \mathcal{Q}'^x_{lidar}), \\ Y^y_{cam} &= \mathcal{B}_{LG}(\mathcal{Q}^y_{cam}, \mathcal{Q}'^y_{lidar}), \\ \mathcal{Q}'_{cam} &= \mathcal{U}(Y^x_{cam}, \pi_x) + \mathcal{U}(Y^y_{cam}, \pi_y), \end{aligned} \tag{14}$$

where $\mathcal{U}$ is the un-shuffling operator.

The complete LG-Mamba module is constructed by stacking multiple LG-Mamba layers for progressive fusion. The refined camera queries from one layer serve as the input to the next. The initial refined LiDAR queries $\mathcal{Q}'_{lidar}$ act as a constant guidance signal, allowing the camera queries to become increasingly imbued with accurate structural awareness while retaining the rich semantics.

## 4 EXPERIMENTS

### 4.1 EXPERIMENTAL SETUP

**Dataset and Metrics.** We conduct all experiments on the nuScenes dataset (Caesar et al., 2020), a large-scale benchmark for autonomous driving. It contains 1000 driving scenes, split into 700 for training, 150 for validation, and 150 for testing. Each sample includes data from a 32-beam LiDAR and 6 surround-view cameras, with annotations for 10 object categories. We follow the official evaluation protocol, using the mean Average Precision (mAP) and the nuScenes Detection Score (NDS) as our primary metrics.

**Architecture.** Our framework builds upon the FUTR3D (Chen et al., 2023) architecture within the MMDetection3D codebase (Contributors, 2020). For feature extraction, it employs HEDNet (Zhang et al., 2023) sparse convolutional backbone for LiDAR and VoVNet (Lee et al., 2019) for

Table 2: Performance comparison of different LiDAR query interactors with two distinct BEV feature encoders.

| Encoder | Interactor | mAP↑ | NDS↑ | FPS |
|---------|-----------|------|------|-----|
| FUTR3D | Attention | 63.7 | 69.1 | 9.3 |
|  | Bi-Mamba | 63.3 | 68.7 | **9.5** |
|  | SA-Mamba | **64.8** | **70.0** | 9.4 |
| HEDNet | Attention | 66.6 | 71.0 | 11.1 |
|  | Bi-Mamba | 66.5 | 70.9 | **11.4** |
|  | SA-Mamba | **67.0** | **71.3** | 11.2 |

Table 3: Ablation study of the core components, SA-Mamba and LG-Mamba, on the nuScenes validation set.

| Backbone | SA-Mamba | LG-Mamba | mAP↑ | NDS↑ |
|----------|----------|----------|------|------|
| ResNet | ✗ | ✗ | 67.4 | 70.9 |
|  | ✓ | ✗ | 68.3 | 71.9 |
|  | ✗ | ✓ | 68.6 | 72.1 |
|  | ✓ | ✓ | **69.9** | **72.6** |
| VoVNet | ✗ | ✗ | 70.3 | 73.1 |
|  | ✓ | ✗ | 70.5 | 73.2 |
|  | ✗ | ✓ | 70.7 | 73.5 |
|  | ✓ | ✓ | **71.1** | **73.7** |

camera inputs. The detection head is a 6-layer decoder processing 900 learnable queries. Following FUTR3D (Chen et al., 2023), during the training stage, we incorporate an auxiliary CenterPoint-based head (Yin et al., 2021) for denser supervision, which is removed at inference.

**Training Strategy.** We adopt the same loss function as FUTR3D (Chen et al., 2023), which consists of a primary detection loss $\mathcal{L}_{det}$ and an auxiliary head loss $\mathcal{L}_{aux}$. The primary loss is a weighted sum of the focal loss (Lin et al., 2017) for classification (with weight $\lambda_{cls} = 2.0$) and the L1 loss for bounding box regression (with weight $\lambda_{bbox} = 0.25$). The auxiliary loss, based on CenterPoint (Yin et al., 2021), provides dense supervision. We employ a two-stage training strategy on a server with 8 NVIDIA RTX 4090 GPUs. The process begins

Table 4: Analysis of layer stack for SA-Mamba and LG-Mamba. Latency is broken down into core computation ($T_c$) and Z-order overhead ($T_s$), all in ms.

| Module | Layers | $T_c$ | $T_s$ | mAP↑ | NDS↑ |
|--------|--------|-------|-------|------|------|
| SA-Mamba | 1 | 0.85 | 0.27 | 66.8 | 71.1 |
|  | 2 | 1.70 | 0.53 | **67.0** | **71.3** |
| LG-Mamba | 1 | 1.32 | - | 70.9 | 73.6 |
|  | 2 | 2.57 | - | **71.1** | **73.7** |

by pre-training a LiDAR-only model incorporating our SA-Mamba module for 20 epochs. This pre-training leverages Class-Balanced Grouping and Sampling (CBGS) (Zhu et al., 2019) to address class imbalance and is optimized with AdamW under a one-cycle learning rate schedule (cycling between $1 \times 10^{-4}$ and $1 \times 10^{-3}$). Subsequently, we transition to multi-modal training for an additional 6 epochs, where we initialize the LiDAR branch from our pre-trained checkpoint and the camera branch from DETR3D (Wang et al., 2021). During this stage, we freeze the backbones while training only the fusion mechanism. This fine-tuning stage uses AdamW with a base learning rate of $2 \times 10^{-4}$ and a cosine annealing schedule. For both two stages, we set the batch size as 16 with 0.01 weight decay.

## 4.2 MAIN RESULTS

We evaluate the overall performance of our S3M3D framework on the nuScenes benchmark and compare it with state-of-the-art methods. Table 1 reports the quantitative results.

**LiDAR-only Performance.** We first present the results of our LiDAR-only model, S3M3D-L. In this configuration, we replace the standard self-attention layers in the FUTR3D-L baseline with our SA-Mamba module, achieving 67.8% mAP and 71.8% NDS (Table 1). Such gains correspond to a 1.9% improvement in NDS over the FUTR3D-L baseline. These results demonstrate that explicitly modeling geometric priors enables SA-Mamba to provide a stronger representation for LiDAR queries than standard self-attention, thereby laying a solid foundation for multi-modal fusion.

**Multi-modal Performance.** The full S3M3D model, integrating both LiDAR and camera data, demonstrates the effectiveness of our spatially-aware representation and guidance-based fusion mechanism. To ensure a fair and direct comparison with the FUTR3D baseline, we employ the same VoVNet backbone for the camera. As shown in Table 1, our S3M3D model achieves 71.2% mAP and 73.6% NDS on the nuScenes test set. This result not only exceeds the FUTR3D baseline by 1.5% NDS, but also outperforms other leading methods such as BEVFusion and TransFusion. The performance improvement is noteworthy, especially considering that our training strategy freezes the backbones. This directly attributes the gains to our novel detection head design, where geometrically-aware representations from SA-Mamba effectively guide camera feature processing

via LG-Mamba. These results validate that our asymmetric, guidance-based fusion paradigm is more effective than the symmetric approaches in previous works.

## 4.3 ABLATION STUDIES

In this section, we conduct a series of ablation studies on the nuScenes validation set to evaluate the effectiveness of our proposed components. We systematically analyze the individual contributions of SA-Mamba and LG-Mamba, their synergy, and the impact of key design parameters on the trade-off between performance and efficiency. All experiments are conducted on the FUTR3D framework unless otherwise specified.

**Effectiveness and Efficiency of SA-Mamba.** We begin by analyzing the SA-Mamba module in a LiDAR-only setting. As shown in Table 2, SA-Mamba consistently outperforms the self-attention baseline, yielding NDS gains up to 0.9%, whereas a naive bidirectional Mamba without spatial ordering degrades performance. Table 4 further shows that this accuracy is achieved at a comparable inference speed. The Z-order permutation computation overhead ($T_s$) is minimal, making the module's total latency negligible relative to the overall detection pipeline. These results highlight SA-Mamba's practical value, offering a substantial performance improvement with minimal computational overhead.

**Component Contributions and Synergy.** We next analyze the multi-modal components and their synergy, assessing the individual and joint contributions of SA-Mamba and LG-Mamba across backbones (Table 3). Individually, both SA-Mamba and LG-Mamba improve upon the FUTR3D baseline. With the VoVNet backbone, adding only SA-Mamba boosts mAP by 0.2%, whereas incorporating only LG-Mamba yields a larger gain of 0.4%. This underscores the important role of our asymmetric guidance mechanism in resolving cross-modal ambiguities. The full S3M3D model, which combines both modules, achieves the best performance on both backbones, reaching 72.6% NDS with ResNet101 and 73.7% NDS with VoVNet. This indicates a clear synergistic effect: the spatially-aware representations from SA-Mamba serve as a higher-quality conditioning signal, enhancing the effectiveness of LG-Mamba's guidance mechanism in refining camera-based queries.

**Backbone Sensitivity and Layer Depth.** We next probe when the observed synergy is most beneficial and how model capacity should be allocated. To this end, we vary the camera backbone and the layer depth (Tables 3 and 4). With backbone variation, the performance uplift is more pronounced on the weaker ResNet101 backbone (+1.7% NDS) than on the stronger VoVNet backbone (+0.6% NDS). This finding supports our core hypothesis: when the camera backbone is less powerful, its features are more susceptible to spatial ambiguity and projection noise, and our asymmetric guidance is most effective in this setting. The robust, geometrically-aware LiDAR representations from SA-Mamba provide a structural anchor, allowing LG-Mamba to better constrain and refine the less reliable camera queries, underscoring the robustness and practical value of the design. We then examine module capacity by varying the layer depth (Table 4). For SA-Mamba, increasing the layers from one to two yields an additional 0.2% NDS, indicating that iterative spatial refinement is beneficial. A similar trend is observed for LG-Mamba, where stacking a second layer improves mAP from 70.9% to 71.1%. In both cases, the gains come with a moderate increase in latency. We thus adopt a 2-layer configuration for both SA-Mamba and LG-Mamba in the final model as a good balance between performance and efficiency.

## 5 CONCLUSION

This paper introduces S3M3D, a novel framework that resolves the symmetric fusion bottleneck in multi-modal 3D object detection by abandoning the flawed assumption of equal reliability between LiDAR and camera queries. Our approach features two synergistic and Mamba-based components: SA-Mamba and LG-Mamba. The SA-Mamba replaces self-attention to encode crucial geometric priors into LiDAR queries by processing them along a recursive Z-order curve. The LG-Mamba establishes an asymmetric mechanism where these refined LiDAR queries guide the state-space processing of less reliable camera queries. Extensive experiments on the nuScenes benchmark validate our approach, demonstrating that S3M3D achieves state-of-the-art performance by establishing a more robust and geometrically-grounded perception paradigm through this principled and guidance-based fusion.

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
