# A APPENDIX

## A.1 ANALYSIS OF DETECTION ERRORS

As shown in Table 5, our method reduces four of the five key error metrics, including translation, scale, velocity, and attributes. These gains stem from the enhanced geometric precision of SA-Mamba and the cross-modal noise suppression of LG-Mamba, validating our design's effectiveness in improving overall localization and size estimation.

Table 5: Comparison of error metrics on the nuScenes validation set. S3M3D-L replaces self-attention with a two-layer SA-Mamba, while the full S3M3D model additionally incorporates a two-layer LG-Mamba for asymmetric modulation.

| Method | mATE ↓ | mASE ↓ | mAOE ↓ | mAVE ↓ | mAAE ↓ |
|--------|--------|--------|--------|--------|--------|
| FUTR3D-L | 31.3 | 26.4 | 26.4 | 24.5 | 19.0 |
| S3M3D-L | **30.4** | **25.2** | **25.3** | **23.2** | **18.0** |
| FUTR3D | 31.2 | 25.0 | **19.4** | 25.8 | 18.3 |
| S3M3D | **30.3** | **24.1** | 25.1 | **24.3** | **17.9** |

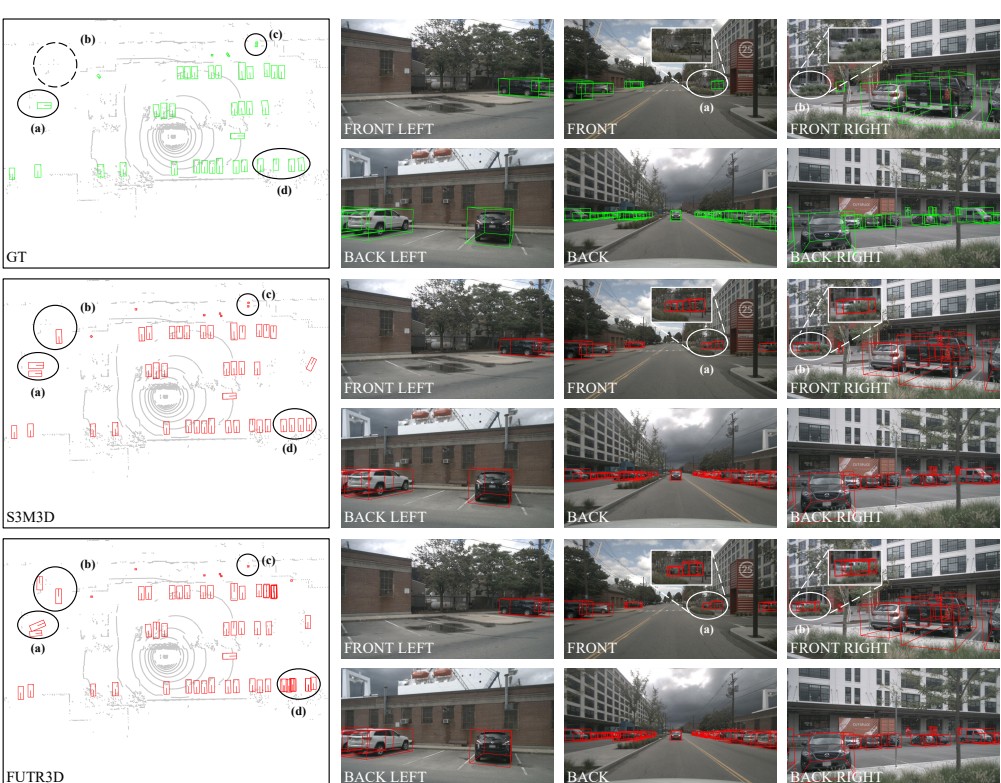

Figure 5: Comparison against FUTR3D on the nuScenes validation set (GT in green, predictions in red). S3M3D shows superior performance in challenging cases, while the baseline model often fails or produces inaccurate predictions.

## A.2 QUALITATIVE ANALYSIS

Figure 5 provides a qualitative comparison that illustrates S3M3D's superior detection capabilities over the FUTR3D baseline in challenging real-world scenarios. Our model exhibits enhanced robustness, delivering more accurate heading angle estimations for unannotated objects (a) and avoiding the hallucinatory false positives that FUTR3D occasionally produces (b), demonstrating a better grasp of scene geometry and improved reliability. Furthermore, S3M3D proves more effective in complex scenes, successfully identifying distant pedestrians in cluttered environments where the baseline model fails (c), showcasing greater sensitivity to small or faraway targets. This improved performance is significantly aided by our architectural design: the SA-Mamba module's recursive z-order scan on the BEV plane effectively resolves spatial ambiguities, thereby reducing detection overlap among densely parked vehicles (d) and leading to cleaner, more reliable results.