# OpenReview forum: "S3M3D: Cross-Modal Selective State Space Modulation for 3D Object Detection"
_ICLR.cc/2026/Conference — ICLR 2026 Conference Withdrawn Submission_

### Official Review · Reviewer_t5wv · 2025-10-26

**Soundness:** 3
**Presentation:** 3
**Contribution:** 3
**Rating:** 6
**Confidence:** 3

**Summary:**

In this paper, the author first claim the problem exists in previous transformer-based 3D object detection models. The permutation invariance in attention will prevent the model from capturing spatial arrangement and geometric prior and the equal treatment of camera and LiDAR queries that have imbalanced reliability will lower the effectiveness of LiDAR. Therefore, the authors propose two new mechanisms that are SA-Mamba and asymmetric LiDAR guied inter-modality fusion to copy with the two problems separately. SA-Mamba implements orders in queries and LiDAR guided fusion increase the effectiveness of LiDAR. The experiments show the effectiveness of the proposed method.

**Strengths:**

1. The writing is easy to follow.

2. The proposed method effectively improves the performance of backbone model.

3. The ablation studies show the effect of both proposed mechanisms.

**Weaknesses:**

1. All experiments are based on FUTR3D backbone. To show the generalization effect of the proposed method, could you provide experiment results when using different backbone?

2. Could you please explain how the asymmetric LiDAR guided fusion works more detaily? Because from the Figure 4 and equations, it seems that the fusion process is still symmetric.

**Questions:**

Please refer to Weakness section

---

### Official Review · Reviewer_wZQV · 2025-10-26

**Soundness:** 2
**Presentation:** 2
**Contribution:** 2
**Rating:** 2
**Confidence:** 4

**Summary:**

This paper proposes S3M3D, a novel framework for multi-modal 3D object detection that addresses the symmetric fusion bottleneck in existing query-based methods. The key insight is that current approaches treat geometrically precise LiDAR queries and uncertain camera queries with equal reliability, missing the opportunity to leverage high-fidelity LiDAR queries to guide noisy camera queries. The authors introduce two synergistic Mamba-based components: (1) Spatially-Aware Mamba (SA-Mamba) that models LiDAR query interactions using recursive Z-order serialization to capture geometric priors, and (2) LiDAR-Guided Mamba (LG-Mamba) that establishes asymmetric guidance where robust LiDAR queries modulate the state-space processing of camera queries. Experiments on nuScenes demonstrate state-of-the-art performance.

**Strengths:**

1) The method achieves competitive performance on nuScenes, with consistent improvements across different backbones and comprehensive ablation studies validating each component's contribution.
2) The authors demonstrate that their approach maintains reasonable computational overhead while achieving performance gains, making it practically viable.

**Weaknesses:**

1) The performance of S3M3D is not SOTA although the authors use advanced LiDAR network HEDNet and image network VoVNet. the paper fails to compare against recent multi-modal fusion methods from 2024-2025.
2) The paper lacks rigorous theoretical justification for key design choices. Most notably, there is no analysis of why the Z-order curve is optimal compared to other space-filling curves like Hilbert or Morton curves. The authors provide no theoretical or empirical comparison of different serialization strategies. Additionally, the paper lacks analysis of the selective SSM parameters (Δ, B, C), their evolution during training, convergence properties, or stability of the proposed formulation. This theoretical gap undermines the scientific rigor of the contribution.
3) The paper fails to provide adequate analysis of failure cases, robustness to sensor degradation, or performance under challenging conditions. Given that the method relies heavily on LiDAR guidance, there is no discussion of what happens when LiDAR data is sparse, corrupted, or missing entirely.

**Questions:**

1.What happens when LiDAR data is corrupted or missing? Does the asymmetric design make the system more fragile?
2.Have you experimented with other space-filling curves beyond Z-order?
3.How does the method scale to larger scenes or higher resolution inputs?

---

### Official Review · Reviewer_XPTQ · 2025-10-29

**Soundness:** 2
**Presentation:** 2
**Contribution:** 2
**Rating:** 2
**Confidence:** 4

**Summary:**

Multi-modal 3D object detection is critical for autonomous driving. However, prevailing query-based methods suffer from a symmetric fusion bottleneck, treating geometrically precise LiDAR queries and uncertain camera queries with equal reliability. To address the above problem, the authors proposed a 3D object detector method named as S3M3D. Spatially-Aware Mamba (SA-Mamba) is designed to model interactions among LiDAR queries, replacing self-attention. LiDAR-Guided Mamba (LG-Mamba) is then proposed to establish an asymmetric guidance mechanism to avoid the less reliable camera queries. Experiments on the nuScenes dataset show the effectivness of the proposed method.

**Strengths:**

1. The structure is well-written and the pipeline is easy to understand.
2. Figs. 1-3 are insightful which convey the novelty to the readers.
3. Although experiments are compared with SoTA methods, there lacks experiments on the other public datasets.

**Weaknesses:**

1. Lack of explicit computational and runtime comparison.
A central motivation for adopting State Space Models (SSMs) such as Mamba is their linear-time complexity and claimed efficiency advantage over the quadratic complexity of self-attention. However, the paper does not provide a detailed computational analysis comparing S3M3D with Transformer-based baselines like FUTR3D. Beyond Table 2, there is no explicit report of FLOPs, inference latency, or training time. While the proposed model achieves higher accuracy, a discussion on efficiency trade-offs is critical—particularly for real-time domains such as autonomous driving. The authors should include explicit runtime comparisons (e.g., FPS per modality, total end-to-end latency) to substantiate claims of efficiency and demonstrate the practical viability of S3M3D relative to attention-based counterparts.

2. Missing ablations on design choices.
The proposed use of recursive Z-order serialization and asymmetric LiDAR-guided fusion are key innovations, yet their design space is not sufficiently explored. Specifically:
A comparison between Z-order, Hilbert curve, and random serialization would clarify whether the performance gain stems from spatial ordering itself or the particular choice of Z-order indexing.
An ablation contrasting asymmetric (LiDAR→Camera) vs symmetric fusion is also necessary to isolate the contribution of the proposed guidance mechanism.
Such analyses would strengthen the empirical justification for the architecture design and demonstrate robustness to implementation variants.

3. Lack of theoretical justification.
While the engineering design of SA-Mamba and LG-Mamba is elegant and empirically effective, the theoretical reasoning behind their superiority remains underexplored. The paper claims that Mamba better captures geometric priors compared to self-attention, but this assertion is demonstrated only through performance metrics. A deeper explanation—perhaps linking the recursive Z-order scan and the inductive bias of SSM dynamics to geometric structure modeling—would considerably improve the conceptual depth and interpretability of the work.

4. Clarification on the self-attention baseline.
The paper argues that vanilla self-attention is “structure-agnostic” due to permutation invariance. However, standard Transformer-based 3D detectors (e.g., FUTR3D, DETR3D) typically employ positional encodings derived from 3D reference points to mitigate this issue. It is unclear whether the “Attention” baseline in Table 2 includes such positional encodings.
If positional encodings were omitted, the comparison may underestimate the performance of the self-attention baseline, making it unfairly weak.
If they were included, then the claim that self-attention is completely structure-agnostic should be nuanced, emphasizing that Mamba captures long-range geometric dependencies more effectively rather than being the sole structure-aware option.
Clarifying this point is important to ensure the validity of the baseline comparison.

5. Potential failure modes of asymmetric guidance.
The strong asymmetric design, where LiDAR features modulate camera processing, raises potential robustness concerns. In cases where LiDAR data are sparse, noisy, or occluded—while the camera has clear visual evidence—the model might incorrectly suppress valid visual-only detections. The authors should discuss whether such cases occur and how the model behaves when LiDAR guidance is unreliable. An analysis of class-wise performance (e.g., for small or visually salient but sparsely scanned objects like pedestrians or bicycles) could provide valuable insight into this potential failure mode.

**Questions:**

It is essential to address the concerns in the weakness.

---

### Official Review · Reviewer_7hCm · 2025-10-30

**Soundness:** 3
**Presentation:** 2
**Contribution:** 2
**Rating:** 4
**Confidence:** 4

**Summary:**

The paper proposes a new method for 3D multi-sensor object detection for autonomous driving, S3M3D. Whereas existing state-of-the-art architectures use attention mechanisms for fusing sensor information, this paper proposes to use a more efficient Mamba-style state-space model (SSM) for fusion. The key challenge is that SSMs are designed for ordered sequential data, but in the BEV object detection paradigm tokens have associated 2D spatial locations that do not have an inherent strict order. S3M3D enables to use of SSMs with two new modules: first, a Spatially-Aware Mamba (SA-Mamba) to processes 2D spatial queries in two orthogonal sequential Z-orders, and second, an asymmetric LiDAR-Guided module to fuse lidar and camera tokens (LG-Mamba). Experiments on nuScenes show that S3M3D outperforms the baseline attention-based architecture, while obtaining even slightly higher frames-per-second (FPS). Ablation studies compare the impact of different Mamba components, backbones, and BEV encoders.

**Strengths:**

* The paper introduces the use of Mamba-style SSMs in the multi-modal sensor fusion layer of a BEV object detection. The general concept appears sound and intuitive.
* Experiments on nuScenes show that this design outperforms the conventional attention mechanism in FUTR3D baseline architecture on which it was based in terms of mAP and NDS.
* Ablation studies using multiple backbones and BEV feature encoders, and show the proposes model has slightly higher FPS (efficiency) than attention.
* Experiments include both lidar-only and lidar+camera performance.

**Weaknesses:**

* Methodological contributions are not so novel or well-motivated:
    * Contribution 1, SA-Mamba address the problem of creating a sequential scanning pattern for spatial geometry. The main approach of combining mutiple scanning pattern directions was already proposed in VMamba ([Liu,NeurIPS'24], cited in paper) and using z-order curves for covering areas/volumes was already proposed in PointMamba ([Liang,NeurIPS'24], not cited) and VoxelMamba ([Zhang,NeurIPS'24], cited). The paper acknowledges this, and explains that the novelty is that these techniques were used for backbones and not for fusion layer (line 135), so this is mostly a novel application of existing techniques.
    * Contribution 2 is an assymetric cross-modal fusion mechanism, departing from symmetric fusion. However, asymmetric multi-sensor fusion is very common, and not novel per se. Examples includes pioneering works as PointPainting [Vora,CVPR'20], and TransFusion [Bai,CVPR'22]. In fact, this asymmetric dependence of one sensor on another is often considered a weakness, as such methods cannot work if one modality is missing or corrupted, which was the motivation for BEVFusion [Liang,NeurIPS'22] and many newer works to pursue symmetric architectures. In fact, Table 1 presumably does not include camera-only performance for this reason. All in all, it is not really clear why asymmetry is considered a positive architecture design by itself.

* Experimental comparison misses state-of-the-art baselines. For example, the paper discusses CMT and DeepInteraction++ in the Related Work, but doesn't include their results in the main comparison table. It is not clear to me why. For instance, [Yin'24] shows test-set L+C mAP of about 73.0, and CMT 72.0, while reporting the same TransFusion results (showing tables are comparable). It is also unclear if the proposed Mamba-fusion strategy could be integrated in such stronger baselines, or only works for the FUTR3D architecture (see also my Questions below).

* Clarity:
	* The concept of "Z-order" is not explained, and assumed familiar to the reader. This can be confusing for readers familiar with more conventional object detections, but not necessarily with Mamba. Originally, I thought Z-order referred to the order of the queries in the spatial z ("vertical"?) dimension. A reference to or short explanation of a Z-curve would have helped.
	* line 193, "The inherent limitation of self-attention in query-based 3D object detectors is that it treats object queries as an unstructured set, ignoring their crucial spatial relationships."; I find this a strange statement: in transformer architectures the tokens include positional embeddings, allowing the attention mechanism to weigh spatial relations between queries and keys. So, this is not an inherent limitation.


[Liang,NeurIPS'24]: Liang, Dingkang, et al. "PointMamba: A simple state space model for point cloud analysis." Advances in neural information processing systems 37 (2024): 32653-32677.

[Yin'24]: Yin, Junbo, et al. "Is-fusion: Instance-scene collaborative fusion for multimodal 3d object detection." Proceedings of the IEEE/CVF conference on computer vision and pattern recognition. 2024.

**Questions:**

* How specific are your Mamba fusion blocks to the FUTR3D architecture? Could they also be integrated to other architectures, e.g. CMT, DeepInteraction++, etc.? Why is there no comparison against those stronger baselines in your main table?

* In Table 3, what is exactly happening in the variant without SA-Mamba and LG-Mamba? Would this be the "Attention" or "Bi-Mamba" baseline from Table 2? Why are the report mAP and NDS then different?

* Why does Table 4, stop at 2 layers? Could it not be that more layers performs even better? And what is the fundamental trade-off here?

---

### Note · Authors · 2025-11-16

I have read and agree with the venue's withdrawal policy on behalf of myself and my co-authors.